# Mitigating Head Position Bias in Perivascular Fluid Imaging: LD-ALPS, a Novel Method for DTI-ALPS Calculation

**DOI:** 10.3390/neurosci6040101

**Published:** 2025-10-07

**Authors:** Ford Burles, Emily Sallis, Daniel C. Kopala-Sibley, Giuseppe Iaria

**Affiliations:** 1Psylinks Neurotech Corp., Calgary, AB T2W 0E1, Canada; 2Canadian Space Health Research Network, Calgary, AB T2N 1N4, Canada; 3Neurolab, Department of Psychology, University of Calgary, Calgary, AB T2N 1N4, Canada; 4Hotchkiss Brain Institute (HBI), University of Calgary, Calgary, AB T2N 4N1, Canada; 5Department of Psychiatry, Cumming School of Medicine, University of Calgary, Calgary, AB T2N 4N1, Canada; 6Alberta Children’s Hospital Research Institute (ACHRI), University of Calgary, Calgary, AB T2N 4N1, Canada; 7Mathison Centre for Mental Health Research and Education, University of Calgary, Calgary, AB T2N 4Z6, Canada

**Keywords:** glymphatic system, diffusion tensor imaging (DTI), neurodegenerative diseases, Alzheimer’s disease

## Abstract

Background/Objectives: The glymphatic system is a recently characterized glial-dependent waste clearance pathway in the brain, which makes use of perivascular spaces for cerebrospinal fluid exchange. Diffusion tensor imaging analysis along the perivascular space (DTI-ALPS) offers a non-invasive method for estimating perivascular flow, but its biological specificity and susceptibility to methodological variation, particularly head position during MRI acquisition, remain as threats to the validity of this technique. This study aimed to assess the prevalence of current DTI-ALPS practices, evaluate the impact of head orientation on ALPS index calculation, and propose a novel computational approach to improve measurement validity. Methods: We briefly reviewed DTI-ALPS literature to determine the use of head-orientation correction strategies. We then analyzed diffusion MRI data from 172 participants in the Alzheimer’s Disease Neuroimaging Initiative (ADNI) to quantify the influence of head orientation on ALPS indices computed using the conventional Unrotated-ALPS, a vecrec-corrected ALPS, and the new LD-ALPS method proposed within. Results: A majority of studies employed Unrotated-ALPS, which does not correct for head orientation. In our sample, Unrotated-ALPS values were significantly associated with absolute head pitch (*r*_169_ = −0.513, *p* < 0.001), indicating systematic bias. This relationship was eliminated using either vecreg or LD-ALPS. Additionally, LD-ALPS showed more sensitivity to cognitive status as measured by Mini-Mental State Examination scores. Conclusions: Correcting for head orientation is essential in DTI-ALPS studies. The LD-ALPS method, while computationally more demanding, improves the reliability and sensitivity of perivascular fluid estimates, supporting its use in future research on aging and neurodegeneration.

## 1. Introduction

The glymphatic system is a recently proposed waste-clearance pathway in the brain that plays a crucial role in maintaining neural homeostasis. In this system, cerebrospinal fluid (CSF) enters the brain along para-arterial spaces, moves through the interstitial space facilitated by aquaporin-4 water channels on astrocytic endfeet, and carries metabolic waste products toward para-venous spaces for removal [1]. This flow aids in the clearance of neurotoxic substances, such as amyloid-beta and tau proteins, which are implicated in the pathogenesis of Alzheimer’s disease [2]. The glymphatic system is thought to be supported by lymphatic vessels located in the dural sinuses and meningeal arteries, contributing to the regulation of CSF and interstitial fluid dynamics [3,4].

Understanding the glymphatic system is important because impairments in this pathway have been implicated in various neurological conditions, including Alzheimer’s disease, Parkinson’s disease, and other forms of dementia [5,6,7]. For instance, decreased glymphatic clearance can lead to the accumulation of amyloid-beta plaques and tau tangles, hallmark features of Alzheimer’s disease [8]. Moreover, the efficiency of the glymphatic system is known to decline with age, potentially explaining the increased susceptibility to neurodegenerative disorders in the elderly population [9]. Sleep disturbances, traumatic brain injuries, and cerebrovascular diseases can also disrupt glymphatic function, exacerbating neurological deficits [10,11,12].

Assessing the function of the glymphatic system is therefore of significant clinical interest [13]. By understanding how this system operates in both health and disease, researchers can develop targeted interventions to enhance waste clearance from the brain [14]. Traditional methods for assessing in vivo glymphatic flow in humans often rely on the use of gadolinium-based contrast agents (GBCAs) in conjunction with MRI. These techniques typically use intrathecal administration of GBCAs and multiple MRI sessions to visualize cerebrospinal fluid movement and glymphatic pathways [15,16]. While providing direct imaging of glymphatic function, these traditional methods are more invasive and carry potential risks associated with contrast agent use, including allergic reactions [17] and gadolinium deposition in brain tissues [18].

As a non-invasive alternative, diffusion tensor imaging analysis along the perivascular space (DTI-ALPS; see Figure 1) has emerged as a technique that may provide indirect insights into glymphatic function by estimating perivascular fluid diffusivity. This method quantifies directional diffusivity in white matter regions near medullary veins, which are hypothesized to align with glymphatic flow pathways [19]. While the ALPS index is often interpreted as a proxy for glymphatic activity, its specificity to glymphatic clearance remains debated [20], and it should not be interpreted as a definitive marker of glymphatic function [21]. Rather, ALPS metrics should be interpreted as primarily reflecting white matter geometry and diffusion properties, as opposed to specifically perivascular or glymphatic flow [22].

The relative ease of use of the DTI-ALPS method, as opposed to gadolinium-based methods, is likely one of the major drivers of its use. However, the typical manner in which the DTI-ALPS index is computed in the literature is highly sensitive to the placement of a participant’s head position during MRI acquisition [24], which is a threat to the validity of this technique. In fact, the original DTI-ALPS technique developed by Taoka and colleagues [24,25] involved a carefully specified diffusion acquisition protocol. Specifically, the authors aligned the MRI slice prescription to the anterior commissure to posterior commissure (AC-PC) line, effectively accounting for individual differences in head pitch during scanning. However, this detail was omitted in the original 2017 report of the technique, but ostensibly performed given it was made explicit in future implementations by the same authors. This alignment procedure ensures that the diffusion tensor components correspond with the brain’s canonical anatomical axes, required for an accurate calculation of the ALPS Index. Follow-up studies [26,27] further demonstrated that performing post hoc rotation of diffusion vectors to account for changes in head orientation improved the reproducibility of this measure, both in datasets which were acquired with AC-PC aligned acquisitions as well as those without. This procedure is referred to as ‘vALPS-index’ and ‘vo-ALPS’, terms that did not appear to be adopted by other authors using the same technique (and in the present paper is referred to as ‘VECREG-ALPS’).

Here, we argue that the importance of the post-acquisition vector registration is understated in the literature, particularly in cases where the diffusion slice prescription is not AC-PC aligned. Specifically, we argue that, if vector registration is not performed, the validity, not merely the reliability, of the ALPS index is threatened by differences in head position, most saliently head pitch, during scanning acquisition. Without proper alignment, the primary diffusion directions may not correspond to the canonical anatomical directions of interest, leading to erroneous interpretations of perivascular diffusivity.

Specifically, the sensitivity of the unrotated-ALPS index to head position relative to diffusion slice prescription arises from a geometric property of the projection of tensor components onto fixed anatomical axes. In ideal alignment, the dominant diffusion directions within the ALPS ROIs align with the canonical X, Y, and Z axes, allowing their directional diffusivities (e.g., Dxx, Dyy, Dzz; see Figure 1) to meaningfully reflect the magnitude of diffusion orthogonal to specific white matter tracts. However, when the head is rotated off-axis, these tract-aligned tensors become obliquely oriented relative to the imaging coordinate system. As a result, the directional diffusivities used in the ALPS formula (e.g., Dyy in projection fibers) can become artificially inflated when the anatomical tract orientation rotates toward the axis of measurement. This leads to a systematic alteration of ALPS values with changing head orientation, even in the absence of any underlying physiological change.

To understand the extent of this issue, we first conducted a literature review to identify the proportion of DTI-ALPS studies that ensure vector registration, by either explicitly setting the MRI slice prescription, or performing post-acquisition vector registration. Then, we performed a post hoc analysis of the Alzheimer’s Disease Neuroimaging Initiative (ADNI) dataset to quantify the degree to which head orientation affects unrotated DTI-ALPS calculations, and the degree to which sensitivity to other clinically relevant metrics (in this case MMSE scores) is affected by different versions of this technique. Finally, we proposed a novel method to compute DTI-ALPS that leverages voxelwise diffusion directions, hereafter referred to as ‘Local Diffusion ALPS’ or LD-ALPS, and examined if it more strongly relates to clinically relevant measures of human cognition than previously adopted methods.

## 2. Materials and Methods

### 2.1. Literature Search

We performed a non-exhaustive search to identify scientific papers utilizing the DTI-ALPS technique. This search was merely intended to provide a reasonably representative sample of research that makes use of the DTI-ALPS technique. We performed a search on PubMed using the query “DTI-ALPS” on 13 September 2024. This search identified 154 items with the original report by Taoka and colleagues the earliest and sole publication from 2017, a single publication in 2020, seven publications in 2021, 24 publications in 2022, 51 in 2023, and 70 at the time of search in 2024, representing a very salient uptick in the use of DTI-ALPS.

### 2.2. Classification

Author E.S., supervised by author F.B., then classified the results of this literature search into four categories based on the nature of the DTI-ALPS technique:‘Original-ALPS’—this group includes research that sufficiently adheres to the original DTI-ALPS specification, which is characterized by an anatomically aligned slice prescription, tensor calculation, and index calculation performed in native space.‘Unrotated-ALPS’—this group includes DTI-ALPS analyses in which the authors did not specify that the slice specification was aligned with the subjects’ anatomical orientation, and make no mention of vector registration or similar techniques.‘VECREG-ALPS’—this group includes DTI-ALPS analyses in which tensor metrics are rotated and registered to a template space with canonical orientation for interpretation.‘Not-Applicable’—this group includes non-primary research or other research not amenable to the classification scheme outlined above.

### 2.3. Participants

We utilized data obtained from the Alzheimer’s Disease Neuroimaging Initiative (ADNI) database (adni.loni.usc.edu) to compare the three ALPS techniques (Unrotated-ALPS, VECREG-ALPS, and LD-ALPS, detailed below). As such, the investigators within the ADNI contributed to the design and implementation of ADNI and/or provided data but did not participate in analysis or writing of this report. A complete listing of ADNI investigators can be found at: http://adni.loni.usc.edu/wp-content/uploads/how_to_apply/ADNI_Acknowledgement_List.pdf.

Data included in our study were collected from 172 individuals (71 females, *M*(*SD*) age = 73.99 (6.77) years; 101 males, *M*(*SD*) age = 74.81 (7.16) years), classified into five groups by ADNI: 41 individuals were Cognitively Normal, 55 individuals at an Early stage of Mild Cognitive Impairment, 32 individuals at a Late stage of Mild Cognitive Impairment, 34 individuals with Alzheimer’s Disease, and 10 individuals with Significant Memory Concerns. We utilized these individuals’ diffusion MRI data, as well as their Mini-Mental State Examination (MMSE) [28] scores.

### 2.4. Diffusion Processing

The following preprocessing steps were followed for all data included in this manuscript:Raw diffusion data were cleaned using *dwidenoise* and *mrdegibbs* from MRtrix3 [29].Data were eddy-current corrected using FSL’s *eddy_cuda10.2* [30,31], which includes motion correction procedures.Tensor model was fit to the eddy-corrected data using FSL’s *dtifit*.These data were then nonlinearly warped to FSL’s JHU-ICBM-FA-1mm template using *fnirt*.

At this point, we calculated the Unrotated-ALPS metric by taking the average Apparent Diffusion Coefficient (ADC) of the glymphatic flow direction in the projection and association ROIs, and dividing it by the average ADC of the non-glymphatic flow direction in the projection and association ROIs (see Figure 1). For the VECREG-ALPS metric, we additionally processed the tensor metrics produced in (3) using *vecreg*, again carrying these rotated metrics along the warps generated in (4), then utilizing the same formula for the Unrotated-ALPS metric. We utilized the same ROIs utilized by Liu and colleagues [23] available at https://github.com/gbarisano/alps/, accessed on 26 November 2024. These ROIs are approximately spherical, each 515 mm^3^ in volume, centered along the X-axis line at MNI Y = −16, Z = 27 (Figure 1B). We did not perform any additional quality control on these data.

### 2.5. LD-ALPS

The Local Diffusion ALPS technique was designed to address some of the outstanding issues with conventional DTI-ALPS computations. First, the aforementioned DTI-ALPS technique assumes that the primary diffusion directions in each region of interest (ROI) are orthogonal between projection and association ROIs, consistent within each ROI, and consistent across lateral ROI pairs. These techniques also assume that the glymphatic flow axis is consistent within and between all ROIs. The present technique, in comparison, computes local, orthogonal sets of diffusion directions, only assuming the glymphatic flow axis to be consistent within, but not between ROIs. We expect this technique to be more sensitive to associations with behavior or other metrics, as voxelwise heterogeneity is accounted for as opposed to averaged over. The proposed LD-ALPS processing involves the following steps:We inverted the warps computed in preprocessing (4) and moved the predefined ALPS ROIs into native spaceWithin each of the native-space ROIs, we clustered the primary diffusion vector (i.e., the V1 vectors) using the DBSCAN algorithm [32], with ε = 0.5, min_samples = 5–20. Outlier vectors with great-circle distances exceeding z = 3.5 from the primary cluster center are excluded to ensure primary diffusion direction consistency within each ROI.For each ROI, we select the primary diffusion direction of the voxel passed from (2) with the smallest distance to the cluster center to represent that ROI’s primary diffusion direction.For each passed voxel in each ROI, we utilize that voxel’s primary diffusion direction, and the median diffusion direction computed from (3) from the neighboring ROI (i.e., the median primary diffusion direction from the Right Association ROI would be used for voxels in the Right Projection ROI), to identify the orthogonal vector to these two vectors, representing the axis of glymphatic flow.We compute the apparent diffusion coefficient (ADC) of each of these directions for each passed voxel, using a Clough-Tocher 2D interpolator from a half-sphere orthographic projection of the eddy-corrected vector’s ADCs. To handle half-shell diffusion acquisitions, we presume that the diffusion values are equivalent for the polarity-flipped vectors to allow for continuous interpolation the entire sphere.We compute the mean ADC from each ROI for each of the component ALPS values.These values are used to compute the LD-ALPS index using the typical ALPS formula (Figure 1).

### 2.6. Technique Comparison

Using the ADNI data, we then compared the Unrotated-ALPS, VECREG-ALPS, and LD-ALPS metrics, and their associations with variables of no interest (i.e., head pitch, yaw, and roll) as well as their association with MMSE scores and more traditional diffusion metrics, i.e., Fractional Anisotropy (FA) and Mean Diffusivity (MD).

## 3. Results

### 3.1. Literature Review

Of the 154 manuscripts identified in our search (See Appendix A), 141 were research papers reporting the use of a DTI-ALPS technique. Of those, 11 reported using the ‘Original-ALPS’ technique [24,33,34,35,36,37,38,39,40,41,42] with an anatomically aligned slice prescription (note the original ALPS paper did not report this and was not included in this count), eight reported using vecreg [26,27,43,44,45,46,47,48], and were classified as using the ‘VECREG-ALPS’ approach. In the remaining 121, we did not identify the mention of aligned slice prescriptions or utilizing vecreg, and thus potentially utilized ‘Unrotated-ALPS’. This suggests that the overwhelming majority (>85%) of implementations of the ALPS technique in the literature may be affected by participant variability in head orientation during MRI data acquisition. Here, we utilized data collected from ADNI to compare how vulnerable the Unrotated-ALPS, VECREG-ALPS, and LD-ALPS metrics are to the validity threat presented by differences in head orientation.

### 3.2. Head Orientation

Participants displayed a range of head orientations from their diffusion acquisitions (Figure 2). Head pitch (*M*(*SD*) = −11.64(7.89)°) varied between subjects most, with differences in head roll (*M*(*SD*) = −0.07(2.23)°) and yaw (*M*(*SD*) = −0.35(3.34)°) generally more constrained. Interestingly, males and females in this sample displayed differences in head position, with males displaying a more chin-raised head pitch relative to females (males *M*(*SD*) = −12.90(7.02)°, females *M*(*SD*) = −9.84(8.72)°, *t*_170_ = 2.542, *p* = 0.012), while head roll (*t*_170_ = 0.048, *p* = 0.962) and yaw (*t*_170_ = −0.611, *p* = 0.542) were similar. These metrics provide some characterization of the expected degree of head orientation variability when collecting MRI data in older adults.

### 3.3. ALPS Indices

ALPS indices computed using the Unrotated-ALPS, VECREG-ALPS, and LD-ALPS are reported in Figure 2B. We excluded a single male subject from all subsequent analyses, as their computed Unrotated and VECREG-ALPS scores were outliers (z of 3.85 and 4.02, respectively). Unrotated-ALPS scores (*M*(*SD*) = 1.26(0.17)) were significantly (*t*_170_ = −6.225, *p* < 0.001) lower than VECREG-ALPS scores (*M*(*SD*) = 1.30(0.13)), which were significantly (*t*_170_ = −15.191, *p* < 0.001) lower than LD-ALPS scores (*M*(*SD*) = 1.43(0.16)). Unrotated-ALPS indices were most strongly correlated with VECREG-ALPS scores (*r*_169_ = 0.816, *p* < 0.001) and to a lesser extent with LD-ALPS scores (*r*_169_ = 0.703, *p* < 0.001). LD-ALPS displayed a relatively stronger association with VECREG-ALPS scores (*r*_169_ = 0.744, *p* < 0.001).

Males displayed significantly lower Unrotated-ALPS (males *M*(*SD*) = 1.22(0.16), females *M*(*SD*) = 1.31(0.17), *t*_169_ = 3.475, *p* < 0.001), VECREG-ALPS (males *M*(*SD*) = 1.29(0.14), females *M*(*SD*) = 1.33(0.12), *t*_169_ = 2.148, *p* = 0.033), and LD-ALPS scores (males *M*(*SD*) = 1.40(0.16), females *M*(*SD*) = 1.47(0.16), *t*_169_ = 2.913, *p* = 0.004) compared to females.

### 3.4. ALPS Associations with Head Orientation

The Unrotated-ALPS index displayed a strong association with head pitch (*r*_169_ = 0.460, *p* < 0.001) as well as a stronger association with absolute head pitch (*r*_169_ = −0.513, *p* < 0.001; see Figure 3), demonstrating that this method is clearly and strongly affected by head orientation during data acquisition. In comparison, the VECREG-ALPS displayed no association with pitch (*r*_169_ = 0.023, *p* = 0.769) or with absolute pitch (*r*_169_ = −0.030, *p* = 0.694), and LD-ALPS similarly did not show an association with pitch (*r*_169_ = 0.080, *p* = 0.298) nor absolute pitch (*r*_169_ = −0.095, *p* = 0.217). No indices displayed significant associations with raw or absolute yaw or roll (*p*s ≥ 0.129).

### 3.5. ALPS Associations with MMSE

ALPS indices generally showed weak but significant associations (*p*s < 0.001) with cognitive functioning as measured by the MMSE (See Figure 4): Unrotated-ALPS scores showed the relatively weakest association with MMSE, at *r*_169_ = 0.250, followed by nominally stronger VECREG-ALPS at *r*_169_ = 0.260, and finally LD-ALPS showing the strongest association at *r*_169_ = 0.310. Accounting for absolute head pitch using a partial correlation, VECREG-ALPS scores shows the weakest association with MMSE, at *r*_168_ = 0.260, followed by a Unrotated-ALPS at *r*_168_ = 0.286, and LD-ALPS remained unaffected and displayed the strongest association (*r*_168_ = 0.310).

Of note, the average mean diffusivity (MD) extracted from the ALPS ROIs showed a somewhat stronger association with MMSE scores (*r*_169_ = −0.324, *p* < 0.001). Average fractional anisotropy extracted from the same ROIs did not display a significant association with MMSE (*r*_169_ = 0.090, *p* = 0.241). Given the stronger association seen between MD and MMSE as compared to any ALPS score and MMSE, we computed the partial correlation coefficients between the different ALPS scores and MMSE, conditioning on MD, to identify whether the ALPS metrics provide any additional explanatory power over that explained by MD. Neither the Unrotated-ALPS (*r*_168_ = 0.112, *p* = 0.145) nor VECREG-ALPS (*r*_168_ = 0.117, *p* = 0.129) displayed a significant association with MMSE after correcting for MD, whereas LD-ALPS’ relationship with MMSE remained statistically significant (*r*_168_ = 0.172, *p* = 0.025).

## 4. Discussion

The assessment of the glymphatic system has become increasingly important due to its potential role in neurodegenerative diseases and other neurological conditions [7,13]. Glymphatic imaging offers valuable insights into how waste clearance pathways in the brain may contribute to conditions like Alzheimer’s disease [49,50]. Among the various imaging techniques, the DTI-ALPS method stands out for its non-invasive nature, relative simplicity, and capacity to analyze existing datasets retrospectively. These advantages have led to a notable increase in the adoption of the ALPS technique in recent research, providing an appealing alternative to more invasive methods that require gadolinium-based contrast agents. However, ALPS is not considered a direct measure of glymphatic flow, and caution is warranted in interpreting ALPS indices as direct measures of glymphatic function.

Our literature review revealed that the majority of studies employing the DTI-ALPS technique utilized the Unrotated-ALPS method, an approach that is particularly sensitive to an individual’s head orientation within the scanner. While previous publications have acknowledged the influence of head orientation on ALPS measurements [24,27], the severity and implications of this issue have not been sufficiently emphasized. Our analysis showed a strong association between Unrotated-ALPS scores and absolute head pitch (*r* = −0.513, *p* < 0.001), indicating that variations in head orientation can substantially affect the validity of ALPS index calculations. While we did not detect significant associations with head pitch or roll with the Unrotated-ALPS method, this may be due to the far greater inter-individual variability in head pitch as compared to head yaw or roll in the dataset we analyzed. While we feel that the distribution of head orientation in our work is representative of the type of variability one would expect when collecting DTI data from the general population, we expect that greater differences in head pitch or yaw would indeed influence Unrotated-ALPS scores. We do not feel that the dependence of unrotated-ALPS on head position reflects physiological changes in CSF dynamics, which have been shown with sufficient changes in body position [51], but rather results from geometric projection artifacts due to the misalignment of canonical and anatomical axes used for the ALPS calculation.

It is important to note that, while we classified many studies as using the Unrotated-ALPS technique, including Taoka and colleagues’ original report [25], these studies may indeed have performed aligned slice prescriptions or utilized vector registration methods like vecreg but simply did not report these steps. This possibility underscores a broader perception in the literature that such processes are not critical to mention explicitly. However, our results suggest that these steps are essential for obtaining valid ALPS measurements. Therefore, we strongly recommend that future studies utilizing ALPS indices, at a minimum, adopt and report using vector registration techniques to mitigate head orientation bias.

In our comparative analysis, we introduced the Local Diffusion ALPS (LD-ALPS) technique as a novel method that accounts for regional heterogeneity in diffusion directions. Unlike the other ALPS approaches that assume uniform primary diffusion directions across regions of interest (ROIs), LD-ALPS computes local, orthogonal diffusion directions within each ROI, accounting for voxelwise variability in the directions of “glymphatic” and “non-glymphatic” flow when computing ALPS scores. This approach minimizes biases introduced by head positioning and tract orientation, thereby increasing the robustness of the ALPS index as a measure of perivascular diffusivity. Our results indicate that LD-ALPS shows a slightly improved sensitivity to a clinically relevant measure of cognitive functioning, the Mini-Mental State Examination (MMSE), compared to both the Unrotated and VECREG-ALPS methods. Specifically, LD-ALPS solely maintained a significant association with MMSE scores even after controlling for mean diffusivity (*r* = 0.172, *p* = 0.025), suggesting it captures additional variance related to cognitive impairment that the other ALPS techniques were unable to do. This increased sensitivity, however, comes at the cost of a minor increase in computational complexity, as computing VECREG-ALPS is far simpler than computing LD-ALPS. Although, this additional computational demand is not insignificant, it is a far lower demand than that of the preprocessing pipeline utilized here, and common for most diffusion MRI data. Moreover, the utility of LD-ALPS over the VECREG-ALPS technique requires further validation in additional samples and with other cognitive measures to establish its robustness and generalizability, and this method does not address the validity threats to ALPS as raised by Ringstad [20].

## 5. Conclusions

In conclusion, our study underscores the critical importance of addressing methodological issues in DTI-ALPS [20], particularly head orientation bias [24,27], to ensure reliable interpretations of ALPS indices. While Unrotated-ALPS is widely employed in the literature, our findings suggest this technique is not appropriate to use. At the very least, vector registration techniques (i.e., VECREG-ALPS) should be adopted as a standard practice to mitigate head position bias. The LD-ALPS method presents a promising alternative that offers potential advantages in sensitivity to clinically relevant cognitive measures, although it necessitates further investigation. Future work should focus on validating LD-ALPS across diverse samples, cognitive measures, and neurological conditions, and contextualizing ALPS metrics relative to other known validity threats [22]. By refining these techniques, we can enhance the interpretability and utility of diffusion MRI as a tool for studying perivascular fluid dynamics, ultimately advancing our understanding of brain waste clearance mechanisms and their disruption in neurological disease.

## Figures and Tables

**Figure 1 neurosci-06-00101-f001:**
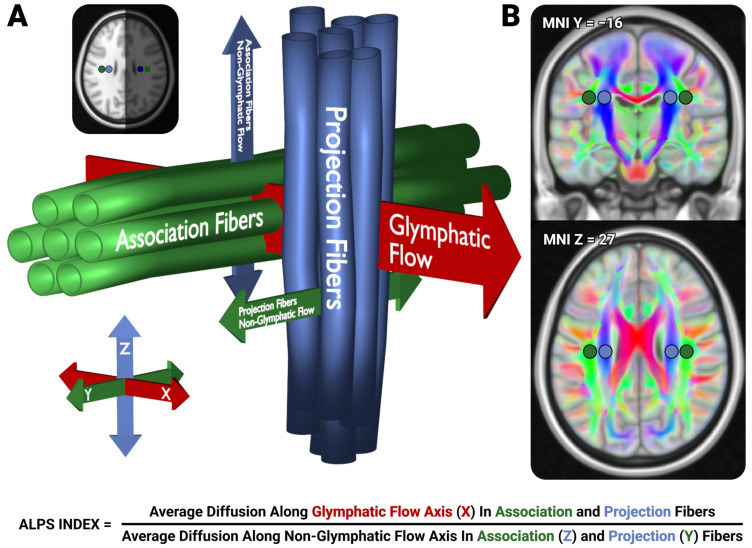
The DTI-ALPS technique. (**A**) depicts the diffusion directions utilized to calculate the ALPS index from the left hemisphere. The projection fibers of the superior coronae radiatae generally travel dorsally–ventrally (i.e., along the Z-axis in blue), whereas the association fibers of the superior longitudinal fasciculi travel anteriorly–posteriorly (i.e., along the Y-axis in green). The direction of glymphatic flow is presumed to be orthogonally traveling laterally (i.e., along the X-axis, in red). (**B**) depicts the location of the spherical ROIs used in the present study, as defined by Liu and colleagues [23]. The ROIs located in the bilateral association fibers (i.e., the superior longitudinal fasciculi) are colored green, whereas the ROIs located in bilateral projection fibers (i.e., the superior coronae radiatae) are colored blue. The ROIs are depicted on a Montreal Neurological Institute (MNI) template brain, overlaid with a depiction of the white matter tracts (from FSL’s standard HCP1065 tensor image) generated by displaying the primary diffusion direction (X-axis in red, Y-axis in green, Z-axis in blue) transparency modulated by the Fractional Anisotropy (FA) value. The ALPS Index is calculated as the average diffusion along the X-axis (i.e., the glymphatic flow axis) in the association and projection ROIs, divided by the average diffusion along the Z-axis in the association ROIs and Y-axis in the projection ROIs (representing the non-glymphatic flow axis).

**Figure 2 neurosci-06-00101-f002:**
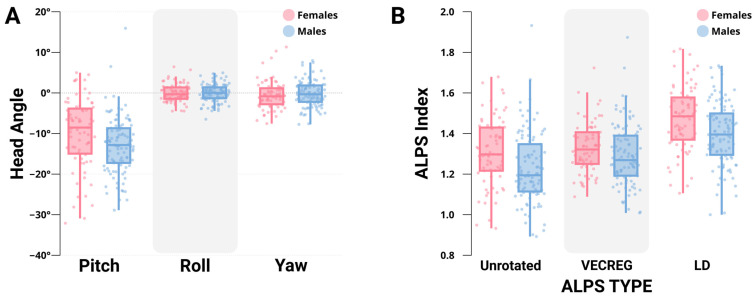
(**A**) depicts the distribution of pitch, roll, and yaw components of male and female participants’ head orientation during DWI acquisition against MNI template space. (**B**) depicts the Unrotated-ALPS, VECREG-ALPS, and LD-ALPS indices in males and females. *N* = 172.

**Figure 3 neurosci-06-00101-f003:**
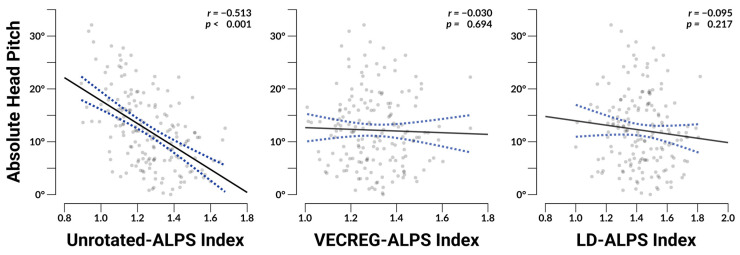
Depicts scatterplots with overlaid regression lines and their 95% confidence intervals indicating the relationship between absolute head pitch and each ALPS index. Reported *r* values represent bivariate Pearson correlation coefficients. N = 171.

**Figure 4 neurosci-06-00101-f004:**
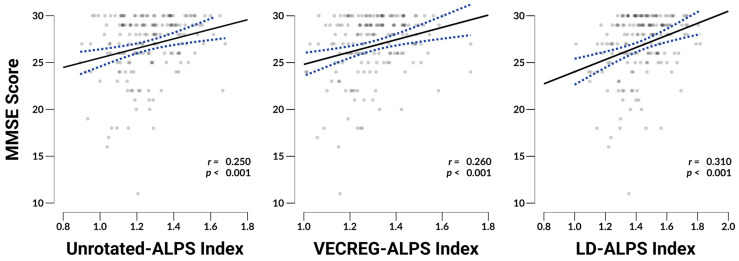
Depicts scatterplots with overlaid regression lines and their 95% confidence intervals indicating the relationship between MMSE and each ALPS index. Reported *r* values represent bivariate Pearson correlation coefficients. N = 171.

## Data Availability

LD-ALPS code will be made available to download at https://fordburles.com/ld-alps.html. Requests for ADNI data should be made to the Alzheimer’s Disease Neuroimaging Initiative.

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
