# Peer review of "Mitigating Head Position Bias in Perivascular Fluid Imaging: LD-ALPS, a Novel Method for DTI-ALPS Calculation"

_neurosci, 2025, doi:10.3390/neurosci6040101_

Round 1
Reviewer 1 Report
Comments and Suggestions for Authors
The article is good and scientifically interesting, here are some minor suggestions for corrections:
- It is desirable to correct some grammatical errors (“it’s/its”, “Alzhaimer’s”, a vecrec-corrected ALPS (p1)).
- Check that all references in the bibliography are actually cited.
- Standardize journal titles and reference formatting.
- For example, reference [22] Liu et al., 2023, indicates “Aging Dis., p. 0.”
- The link “fordburles.com/resources/ld-alps.html” is not available, which is mentioned in the Data Availability Statement.
Author Response
Comment 1: It is desirable to correct some grammatical errors (“it’s/its”, “Alzhaimer’s”, a vecrec-corrected ALPS (p1)).
Response 1: Thank you for noting these errors. I have edited the manuscript and corrected the mistyped 'vecrec', and the misspelled references to Alzheimer's disease, as well as other grammatical errors that ChatGPT identified.
Comment 2: Check that all references in the bibliography are actually cited.
Response 2: I reformatted all references to MDPI style, and ensured that all reference sin the bibliography are cited in-text.
Comment 3: Standardize journal titles and reference formatting (For example, reference [22] Liu et al., 2023, indicates “Aging Dis., p. 0.”)
Response 3: Thank you for identifying this error. I have removed the erroneous reference to page 0 in this reference.
Comment 4: The link “fordburles.com/resources/ld-alps.html” is not available, which is mentioned in the Data Availability Statement.
Response 4: Thank you for raising this issue. I have made this document available at the link provided.
Reviewer 2 Report
Comments and Suggestions for Authors
-
There is no discussion about yaw and roll effect, claiming only pitch matters, may nor generalize.
-
The association between Unrotated-ALPS and head pitch is reported as both positive and negative (e.g., “r169 = −.513” in abstract vs “r169 = .460” in results). Abstract results section has sign inconsistency for correlation. Clarify it.
-
Provide statistical comparisons across multiple diagnostic groups (e.g., ANOVA).
-
The literature search is described as “non-exhaustive” but the methodology is vague.
-
Report the DBSCAN parameters (epsilon, minPts).
-
Provide reason for utilizing Clough-Toucher interpolation vs. other interpolation methods.
-
Provide validation or citation for handling of half-shell acquisition by polarity flipping.
-
Exclusion of one outlier subject is ad hoc (z>3.5); Provide justification for the threshold or whether robust statistics were conducted.
-
Discuss the motion artifacts or quality control beyond denoising and eddy correction.
-
Authors imply that Unrotated-ALPS is invalid; therefore, provide sensitivity analyses to confirm that head alignment or vecreg fully solves the issue across datasets.
-
Abstract contains grammatical issues.
-
Frequent typographical errors are present in the manuscript: “Alzhiemer’s” vs “Alzheimer’s,” “ostensibly” used awkwardly. Avoid long, complex sentences; readability would improve with shorter sentences.
Abstract contains grammatical issues. Frequent typographical errors: “vecrec” vs “vecreg,” “Alzhiemer’s” vs “Alzheimer’s,” “ostensibly” used awkwardly. Long, complex sentences throughout; readability would improve with shorter sentences.
Author Response
Comment 1: There is no discussion about yaw and roll effect, claiming only pitch matters, may nor generalize.
Response 1: Thank you for noting the focus on pitch, as opposed to roll and raw in regards to sensitivity of ALPS metrics to head orientation. We did not detect significant yaw or roll effects (see end of section 3.4), but did note in the discussion (Discussion Paragraph 2) that this may be due to the distribution of head orientation in our dataset in particular, that we feel that this distribution of head orientation is typical of mri studies, and that we believe greater variation of head roll or yaw would indeed affect unrotated ALPS metrics.
Comment 2: The association between Unrotated-ALPS and head pitch is reported as both positive and negative (e.g., “r169 = −.513” in abstract vs “r169 = .460” in results). Abstract results section has sign inconsistency for correlation. Clarify it
Response 2: Thank you for noting this oversight. The statistic in the abstract is now correctly referred to as the association with absolute head pitch, as opposed to simply head pitch. With the omitted 'absolute' addressed, the statistics are now coherent between the abstract and the results.
Comment 3: Provide statistical comparisons across multiple diagnostic groups (e.g., ANOVA)
Response 3: Thank you for suggesting these analyses. I have included them in the supplement.
Comment 4: The literature search is described as “non-exhaustive” but the methodology is vague
Response 4: I edited section 2.1 to more clearly highlight the purpose of the search, and the pubmed query we used. Section 2.2 details the classification performed after the literature search.
Comment 5: Report the DBSCAN parameters (epsilon, minPts)
Response 5: The epsilon and min_samples range are now repored in the LD-ALPS methods section.
Comment 6: Provide reason for utilizing Clough-Toucher interpolation vs. other interpolation methods
Response 6: We utilized a Clough-Tocher interpolation because it is smoother than linear interpolation.
Comment 7: Provide validation or citation for handling of half-shell acquisition by polarity flipping
Reponse 7: I included an explanation for including polarity flipping of half-shell acquisitions. I am not aware of a citation for this procedure, but eigenvectors for tensor calculations used in tractography are assumed to be polarity invariant. Polarity flipping here is quite similar in that it is assuming that these vectors are symmetrical. The ADC distribution is almost certainly not perfectly symmetrical, but we cannot interpolate directions outside of the cases, so polarity flipping ensures that interpolated ADCs are much more likely to approach their true values.
Comment 8: Exclusion of one outlier subject is ad hoc (z>3.5); Provide justification for the threshold or whether robust statistics were conducted
Response 8: The exclusion of the outlier subject was ad hoc, but no threshold was utilized, the exclusion was based off of visual inspection of Figure 2, the extreme z scores were reported merely as a way to quantify the observation. Robust statistics were not conducted, and I ensured that there is no suggestion that we utilized robust statistics in the manuscript.
Comment 9: Discuss the motion artifacts or quality control beyond denoising and eddy correction
Response 9: While processing these data, I did not identify any problematic volumes or egregious motion artifacts, but this was not a formal quality control process. I added text in the methods noting that eddy current correction performs motion correction, and that additional QC procedures were not employed beyond the noted processing.
Comment 10: Authors imply that Unrotated-ALPS is invalid; therefore, provide sensitivity analyses to confirm that head alignment or vecreg fully solves the issue across datasets
Response 10: Thank you for noting the opportunity to more clearly demonstrate that accounting for head alignment does not improve the sensitivity of ALPS scores to clinically-relelvant metrics (i.e., MMSE scores). I have included additional partial correlation analyses in section 3.5 to this effect.
Comment 11: Abstract contains grammatical issues
Response 11: I addressed a handful of grammatical errors in the abstract.
Comment 12: Frequent typographical errors are present in the manuscript: “Alzhiemer’s” vs “Alzheimer’s,” “ostensibly” used awkwardly. Avoid long, complex sentences; readability would improve with shorter sentences
Response 12: Thank you for noting these issues. I have addressed grammatical errors throughout the manuscript.
Reviewer 3 Report
Comments and Suggestions for Authors
The study presents an original and methodologically rigorous contribution through the introduction of the LD-ALPS technique, designed to reduce head position-induced bias in DTI-based ALPS analyses. The results are clearly and effectively presented, and the overall quality of the work is high. However, several aspects warrant refinement prior to publication. The Introduction would benefit from a more comprehensive review of recent and critical literature concerning the validity of the ALPS index and related neuroimaging methodologies, thereby better contextualizing the current study within the evolving research landscape. Furthermore, the Discussion should more thoroughly address the practical implications of LD-ALPS, particularly in terms of its computational requirements and its potential utility in large-scale or clinical imaging studies. While the existing references are largely appropriate, the inclusion of additional up-to-date sources would strengthen the manuscript’s scholarly foundation. These revisions, though minor in scope, would substantially enhance the paper’s impact and academic rigor.
Author Response
Comment 1: The Introduction would benefit from a more comprehensive review of recent and critical literature concerning the validity of the ALPS index and related neuroimaging methodologies, thereby better contextualizing the current study within the evolving research landscape.
Response 1: Thank you for your comments. I agree that a more extensive explanation of the critical literature of ALPS would be beneficial to the reader. I was able to find an additional paper offering a reasoned critique and analysis of ALPS metrics (Schilling et al., 2025) and included it in the introduction. If you have any other citations that you think would be relevant please let me know and I can integrate them into the introduction.
Comment 2: Furthermore, the Discussion should more thoroughly address the practical implications of LD-ALPS, particularly in terms of its computational requirements and its potential utility in large-scale or clinical imaging studies. While the existing references are largely appropriate, the inclusion of additional up-to-date sources would strengthen the manuscript’s scholarly foundation. These revisions, though minor in scope, would substantially enhance the paper’s impact and academic rigor.
Response 2: I agree that expanding upon the vague ‘increased computational requirements’ statement would be valuable for the reader to ascertain if utilizing LD-ALPS is feasible for their project. I more clearly clarified that the demand is higher, but the preprocessing demand (that would be necessary in all cases of analyzing diffusion data) is far higher. Eddy current correction specifically is far more computationally demanding than the LD-ALPS calculation. I intend to add more explicit computational requirements and usage instructions on the website noted in the data availability statement.